# Juvenile Dermatomyositis and Infantile Cerebral Palsy: Aicardi-Gouteres Syndrome, Type 5, with a Novel Mutation in SAMHD1—A Case Report

**DOI:** 10.3390/biomedicines11061693

**Published:** 2023-06-12

**Authors:** Lubov S. Sorokina, Rinat K. Raupov, Mikhail M. Kostik

**Affiliations:** Hospital Pediatry, Saint-Petersburg State Pediatric Medical University, 194100 Saint Petersburg, Russia; lubov.s.sorokina@gmail.com (L.S.S.); rinatraup94@gmail.com (R.K.R.)

**Keywords:** Aicardi-Gouteres syndrome, interferonopathy, SAMHD1, CNS calcifications, sleukodystrophy, infantile cerebral palsy, ICP

## Abstract

Introduction: Aicardi-Gouteres syndrome (AGS) is a monogenic interferonopathy characterized by early onset, dysregulation of skin (chilblain lesions), brain, and immune systems (fever, hepatomegaly, glaucoma, arthritis, myositis, and autoimmune activity). The disease looks like TORCH (Toxoplasmosis, Others, Rubella, Cytomegalovirus, Herpes) infection with early-onset encephalopathy resulting in severe neuropsychological disability. Case description: A six-year-old girl has been suffering from generalized seizures, fever episodes, severe psychomotor development delay, and spastic tetraparesis since the first year of her life. Her two elder brothers died at a young age from suspected infantile cerebral palsy (ICP). Other siblings (younger brother and two elder sisters) are as healthy as their parents. The girl was diagnosed with juvenile dermatomyositis at 5.5 years. Basal ganglia, periventricular, and cerebellum calcifications; hypoplasia of the corpus callosum; and leukodystrophy were detected on CT. The IFN-I score was 12 times higher than normal. The previously not described nucleotide variant c.434G > C (chr 20:36935104C > G; NM_015474) was detected in exon 4 of the SAMHD1 gene in the homozygous state, leading to amino acid substitution p.R145P. Aicardi-Goutières syndrome 5 was diagnosed. Her treatment included corticosteroids, methotrexate, and tofacitinib 5 mg twice a day and it contributed to health improvements. The following brain CT depicted the previously discovered changes without the sign of calcification spreading. Conclusions: Early diagnosis of AGS is highly important as it allows starting treatment in a timely manner. Timely treatment, in return, can help avoid the development/progression of end-organ damage, including severe neurological complications and early death. It is necessary to spread information about AGS among neurologists, neonatologists, infectious disease specialists, and pediatricians. A multidisciplinary team approach is required.

## 1. Introduction

Neurological impairment in children with immune-mediated disease is still actual and difficult to diagnose. Among rheumatic diseases, the central nervous system can be affected by systemic lupus erythematosus, Bechet’s disease, localized scleroderma, Takayasu aortoarteriitis, Kawasaki disease, small-vessels vasculitides, rheumatic fever, and primary CNS vasculitides [1]. There has been rapid progress in the research of autoinflammatory diseases in the last 10–15 years. Aicardi-Gouteres syndrome (AGS) is a monogenic interferonopathy characterized by the early onset and dysregulation of the skin, brain, and immune systems. There are seven subtypes of AGS according to the genes which cause the disease: AGS 1 type (TREX1 gene), AGS 2 type (RNASEH2A gene), AGS 3 type (RNASEH2B gene), AGS 4 type (RNASEH2C), AGS 5 type (SAMHD1 gene), AGS 6 type (ADAR1), and AGS type 7 (IFIH1 gene) [2]. In 2020, when biallelic mutations in LSM11 and RU 7–1 genes had been detected, two new subtypes of AGS syndrome (AGS type 8 and AGS type 9, respectively) were identified. [3]. The most common and typical forms of AGS are AGS type 1 and AGS type 2, which manifest themselves during the first year of life and mimic TORCH (Toxoplasmosis, Others, Rubella, Cytomegalovirus, Herpes) infection. Patients are diagnosed with early-onset encephalopathy resulting in severe neuropsychological disability. The majority of patients with AGS have additional symptoms, including fever, chilblain skin lesions, hepatomegaly, glaucoma, arthritis, myositis, and autoimmune activity. The diagnosis and management of patients with AGS, as well as the treatment protocols, are not yet standardized.

## 2. Case Description

The patient is a 6-year-old girl. She is the fourth child in a large family. Her parents are unrelated and come from the same region. The pregnancy was unremarkable. The birth weight was 3000 g and the body length was 43 cm. Her two elder brothers died at a young age from suspected infantile cerebral palsy (ICP). Other siblings (younger brother and two elder sisters) are as healthy as their parents. Triple X syndrome, ICP, and hip dysplasia were diagnosed in the first year of life. The generalized seizures began at the age of 1.5 months and the patient had been treated with anticonvulsants for five years. There were no seizure relapses during anticonvulsant therapy and after its withdrawal. A low-grade fever of unknown origin was observed during the first year when seizures began. Brain magnetic resonance imaging (MRI) in the first year of life showed ventricular dilatation. The girl has psychomotor development delay and spastic tetraparesis with the 5th level of Gross Motor Function Classification System (GMFCS). She neither talks nor communicates with the outside world and only uses vocalizations and social smiles. Metabolic (tandem mass spectrometry) and mitochondrial disorders (lactate and pyruvate were normal) have been ruled out by genetic counseling.

The erythematous rash appeared on the neck, chest, nape, and face at the age of 5.5 years. Then, periorbital erythema developed. Topical corticosteroids had a partial effect. Wrist edema, periorbital edema, weakness, and low-grade fever were observed during 1 month. Sleep disorder occurred. Infections and allergic conditions were excluded. Juvenile dermatomyositis was suspected and the patient was transferred to the Department of Pediatric Rheumatology. Findings of the examinations were the following: widespread erythematous rash with cyanotic discoloration on the upper extremities, on the chest and the neck, periorbital edema and erythema, and alopecia on the back of the head (Figure 1). The girl had spastic tetraparesis. She also had swelling of her hand joints, as well as swelling and local hyperthermia of the left knee. Laboratory tests revealed mild inflammation (slightly elevated ESR with normal CRP), increased AST = 107 U/L (n.v. < 34 U/L) and LDH = 669 U/L (n.v. < 220U/L) with normal creatine kinase, hypergammaglobulinemia (17.9 g/L, n.v. < 11.2 g/L), and negative immunological markers (antinuclear antibodies and polymyosities antibodies). The whole data are included in Table 1. Whole-body MRI was performed to assess muscle inflammation. Diffuse MR-signal increase on T2 and STIR was determined in the upper and lower muscle extremities and the pelvic and back muscles. Similar changes were found in the fascia and fat tissues. There were signs of bilateral hip joint dislocations. Electroneuromyography revealed signs of an active stage of primary muscular lesion of the pectoralis major and supraspinatus muscles, as well as indirect signs of inactive lesions of the anterior tibial muscle. The clinical manifestations, prior severe neurological impairment, and familial history led to the suspicion of interferonopathy. We performed brain CT, IFN-I score, and clinical exome sequencing. Basal ganglia, periventricular, and cerebellum calcifications; hypoplasia of the corpus callosum; and leukodystrophy were detected on CT (Figure 2). Unfortunately, we did not perform lumbar puncture because of the mother’s rejection. IFN-I scores were 12 times higher than normal. The nucleotide variant c.434G > C (chr 20:36935104C > G; NM_015474) was detected in exon 4 of the SAMHD1 gene in the homozygous state, leading to amino acid substitution p.R145P. This variant has not been described in the gnomAD database v2.1.1 previously. Finally, Aicardi-Goutières syndrome 5 was diagnosed. A short course of corticosteroids (CS) and tofacitinib 5 mg twice a day was started. The rash disappeared. The girl became calmer and began to crawl and roll to the goal. Relapse of rash and myositis were detected in the 8th month of CS tapering (daily dose was <0.2 mg/kg). During the CS tapering, the patient received tofacitinib 5 mg twice a day regularly. The new brain CT depicted the previously discovered changes without a sign of calcification spreading. After confirmation of relapse, we were forced to return to the full dose of CS (1 mg/kg), with a methotrexate treatment as a supplementary steroid-sparing agent, and continue Tofacitinib treatment at the previous dose. Positive dynamics were achieved again. The flow chart of diagnostic procedures and the algorithm is depicted in Figure 3.

## 3. Discussion

In this case report, we presented a rare association of juvenile dermatomyositis and AGS type with new variants in the SAMHD-1 gene.

There are several classic AGS subtypes with prenatal and infantile-onset: spastic-dystonic syndrome, ADAR-1-related bilateral striatal necrosis, hereditary spastic paraparesis, and SAMHD-1-related cerebrovascular disease [4,5,6]. Classical AGS is characterized by TORCH syndrome-mimicking phenotype: microcephaly, abnormal movements, epileptic seizures, thrombocytopenia, anemia, intracranial calcification, and white matter disease on CT. Patients suffer from a progressive disease with early and severe neurologic deficiency. Classic AGS can be presented in all known genotypes. Spastic-dystonic AGS usually has a later onset (after 1st year) than a classic one. It can be similar to classic AGS manifestations or it can have a slowly progressive nervous system injury and variable spasticity and dystonia with normal neuroimaging. Bilateral striatal necrosis is characteristic of ADAR-1 mutations. Spastic paraparesis without spinal or cranial damage can be observed in patients with ADAR1, IFIH1, and RNASEH2B mutations [7].

The first patient with AGS and SAMHD1 mutation was described in 2009 [8]. It was an 8-year-old girl with congenital onset spastic quadriplegia, Moyamoya syndrome, and severe peripheral vasculopathy. Further studies demonstrated that SAMHD1 mutations in AGS patients may act as a negative regulator of the cell-intrinsic antiviral response and lead to interferon type I signaling pathway hyperactivation [9]. Fourteen patients in Old Order Amish with a homozygous mutation in SAMHD1 (c.1411–2A > G) were studied in 2011. They had heterogeneous phenotypes, including normal neurologic and psychomotor development and severe developmental disability. The key features of all patients were cerebral vasculopathy with stenoses and aneurysms in large arteries and skin vasculopathy, such as acrocyanosis, Raynaud’s phenomenon, and chilblain lesions. Glaucoma, arthritis, and migraine headache were observed in some patients [10].

SAMHD1 (sterile alpha motif and HD-domain–containing protein 1) plays a role in double-stranded break repair, genome stability, and the replication stress response through interferon signaling. SAMHD1 has been suggested as a way to down-regulate IFN and inflammatory responses to viral infections [11]. SAMHD1 gene variants lead to inappropriate activation of the IFN-type signaling pathway, following recognition of endogenous DNA and/or RNA species that can activate nucleic-acid-sensing pattern-recognition receptors [12]. It has been hypothesized that this mechanism is not the basis of AGS pathogenesis. A recent study of SAMHD1-deficient mice has demonstrated that increased DNA damage does not result in higher levels of type I interferon. Instead, the chronic interferon response is driven by the MDA5/MAVS pathway [13].

Except for common diagnostic tools that are used in AGS recognition (IFN-α level in cerebrospinal fluid, IFN-1 signature in blood, and clinical genome sequencing) there are specific rare cellular tests that seem to be promising in differential diagnosis for other related conditions (for example, Cockayne and Seckel syndromes) [14,15]. There are several DNA repair proficiency tests in the patient-derived fibroblast cells: transcription-coupled nucleotide excision repair (TC-NER) activity and unscheduled DNA synthesis (UDS) in the patient’s fibroblasts. TC-NER activity, also as UDS, is reduced in AGS patients [16]. It is interesting to note that there was a pilot study (19 patients) with neonatal screening of AGS, which included interferon signature and C26:0 lysophosphatidylcholine level on newborn screening blood spots. The elevation of C26:0 lysophosphatidylcholine was observed in both AGS and X-ALD patients (lower in AGS than in X-ALD individuals; 0.43 μM [0.37–0.48] versus 0.72 μM [0.59–0.84], *p* < 0.001, respectively). At the same time, the concentration was higher in these both conditions than in control observations (0.21 μM [0.21–0.21] *p* < 0.001). Several patients (n = 5) with AGS had negative interferon signatures at birth and, at the same time, RNASEH2B variants [17].

A neurologic features assessment scale has been created for the evaluation of the severity of CNS damage in AGS. It combines gross motor function (GMFCS), communication (CFCS), and manual ability classification systems (MACS) and includes 11 items: normal head size, social smile, vocalizations, possibility to say meaningful words and phrases, head control (>60 s), pincer grasp or self-feeding, independent sitting (>2 min), rolling or crawling to goal, and ambulation with or without assistive devices. The neurologic features assessment scale is correlated with a neurological disability but not with an IFN-1 score.

Our patient described above had a total score of 3/11 before treatment, and 4/11 after it, which means severe neurologic injury. It is important to note that the girl became more active and began to roll and crawl toward the goal on the floor after the treatment initiation (CS and tofacitinib) [18].

### Treatment Strategy

There is no cure for type I interferonopathies. There were attempts to treat AGS with nucleoside reverse transcriptase inhibitors (abacavir, lamivudine, and zidovudine) with a positive effect on the IFN index, which was decreasing; however, there was no neurological improvement [19]. Corticosteroids showed moderate or partial success in the treatment of skin lesions and a decrease in IFN levels in cerebrospinal fluid. However, there were no dynamics of neurological status [20]. This could be explained by the severe irreversible neurological changes at the time of therapy initiation. The most promising targeted treatment is Janus Kinase Inhibitors (JAKI) (tofacitinib, ruxolitinib, and baricitinib) for both monogenic and multifactorial interferon-related diseases that were shown in several scoping reviews [21,22,23,24,25]. The aim of this treatment is to control systemic inflammation and organ-specific disease manifestations and to prevent end-organ damage [26]. JAK inhibitors have improved clinical and analytical parameters and decreased the number of flares, plasma inflammatory markers, and expression of IFN-stimulated genes [21]. At the same time, there are no standardized treatment protocols because the disease is ultra-rare. The next stage in leukodystrophy treatment with promising potential is stem cell therapy and gene therapy via optimizing viral vectors and new techniques, including induced pluripotent stem cells (iPSC) and CRISPR-Cas9 technology [27]. None of them have been used for the treatment of AGS patients yet.

## 4. Conclusions

Early diagnosis of AGS is highly important to start treatment in a timely manner. Timely treatment in return can help to avoid the development/progression of end-organ damage, including severe neurological complications and early death.It is necessary to spread information about AGS among neurologists, neonatologists, infectious disease specialists, and pediatricians.Newborn screening for AGS seems to be a promising tool for early AGS diagnosis.A standardized treatment strategy should be developed according to AGS type and previous experience. A multi-disciplinary team is required to provide optimal care in the context of multiorgan system involvement.

## Figures and Tables

**Figure 1 biomedicines-11-01693-f001:**
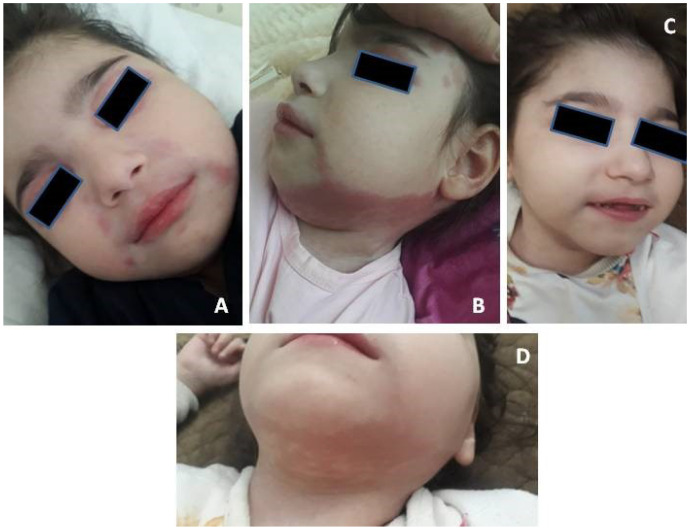
(**A**) Periorbital erythema. (**B**) Erythema on the neck and face. (**C**,**D**) Rash resolution after treatment.

**Figure 2 biomedicines-11-01693-f002:**
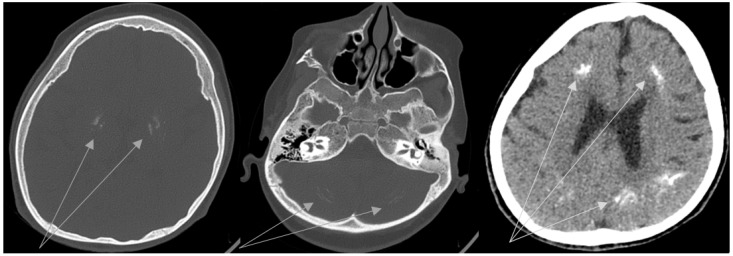
Brain Computed tomography: basal ganglia, periventricular, and cerebellum calcifications (arrows).

**Figure 3 biomedicines-11-01693-f003:**
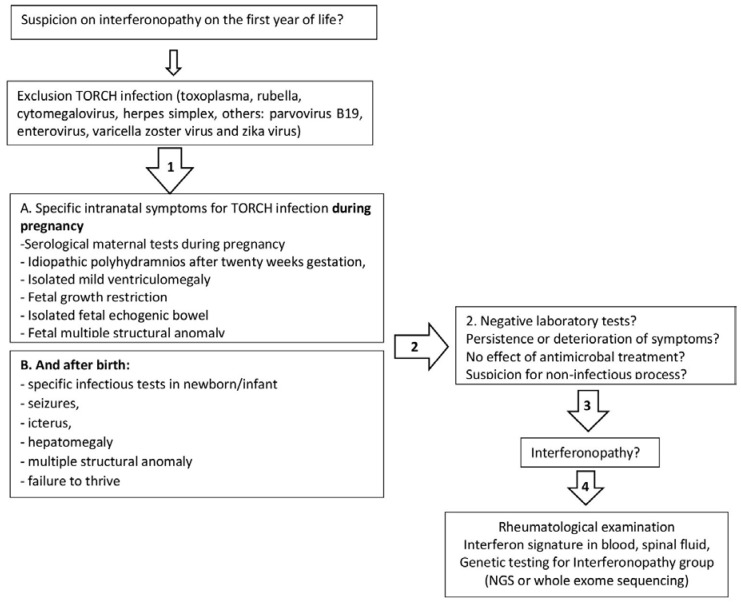
Flow chart of diagnostic procedures and algorithm during disease course.

**Table 1 biomedicines-11-01693-t001:** The laboratorial features of the patient during the diagnosis of the disease.

Parameter	Patient	Normal Value
Hemoglobin, g/L	110	120–160
White blood cells, ×10^9^/L	8.1	4–9
Platelets, ×10^9^/L	318	150–400
Erythrocyte sedimentation rate, mm/h	23	1–20
Fibrinogen, g/L	3.05	2.0–4.0
Lupus anticoagulant *	0.96	0.9–1.2
International normalization ratio	0.97	0.9–1.1
Complement C3, g/L	0.92	0.82–1.73
Complement C4, g/L	0.22	0.13–0.46
Creatine kinase, U/L	90	29–169
**Lactate dehydrogenase, U/L**	669	125–220
Ferritin, ng/mL	115	15–120
Total protein, g/L	69	60–80
Gamma-globulins, g/L	17.5	8.4–11.2
Alanine aminotransferase, U/L	51	0–55
**Aspartate aminotransferase, U/L**	107	5–34
Creatinine, mmol/L	0.047	0.027–0.062
Urea, mmol/L	6.6	2.5–6.0
C-reactive protein, mg/L *	1.9	0–5
Antinuclear antibodies *	1:160	<1:160
Antinuclear antibodies (nRNP/Sm, Sm, SS-A, Ro-52, SS-B, Scl-70, PM-Scl, Jo-1, CENP B, PCNA, dsDNA, Nucleosomes, Histones, Rib P-protein, AMA M2) *	Not detected	Not detected
Polymyositis antibodies (Mi-2, Ku, PM-Scl100, PM-Scl75, SRP, Jo-1, PL-7, PL-12, EJ, OJ, SSA) *	Not detected	Not detected
**Interferon signature, Units**	24	0–2

Footnote: tests, specific for diagnosis are in bold, tests, specific for exclusion are marked with asterisks.

## Data Availability

The original contributions presented in the study are included in the article; further inquiries can be directed to the corresponding author.

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
