# Peer review of "Juvenile Dermatomyositis and Infantile Cerebral Palsy: Aicardi-Gouteres Syndrome, Type 5, with a Novel Mutation in SAMHD1—A Case Report"

_biomedicines, 2023, doi:10.3390/biomedicines11061693_

Round 1

Reviewer 1 Report

The authors report a novel mutation in in SAMHD1  in a patient suffering to juvenile dermatomyositis and infantile cerebral palsy within Aicardi-Gouteres type 5 syndrome in a  6-year-old girl. 

As the authors have stated, early diagnosis of Aicardi-Gouteres syndrome  is of utmost importance for the early treatment to avoid the progression and permanent damage or lethal outcome. 

They aim to raise awareness and advocate for multidisciplinary treatment of these patients.

English language should be revised, while the overall content of the manuscript is satisfactory.

Author Response

Dear Reviewer!

Thank you so much for you kind evaluation of our manuscript.

The English editing has been done now.

On behalf of the Authors

Mikhail Kostik, MD, PhD, Professor

Reviewer 2 Report

The clinical case that the authors present is interesting because it is a new mutation not yet described.

In the presentation of the case, some shortcomings in the description of the diagnosis are highlighted.

For a better description of the clinical case, please add the following data:

- add a diagnostic flow-chart, especially to highlight the diagnoses of exclusions (mimicking TORCH).

- Table 1 shows some laboratory tests. Highlight in bold those specific for prognostic characterization. Highlight with an asterisk those that represent a differential diagnosis of exclusion. Add a legend to the table to describe everything

The role of the laboratory is not clear from the text. Do the tests presented guide and confirm the diagnosis?

If possible, revise the text to improve reading fluency.

Author Response

Dear Reviewer!

Thank you so much for your kind evaluation of our manuscript and valuable suggestions. Our answers (A) on your queries (Q) are below. All changes in the manuscript are highlighted.

The clinical case that the authors present is interesting because it is a new mutation not yet described.

In the presentation of the case, some shortcomings in the description of the diagnosis are highlighted.

For a better description of the clinical case, please add the following data:

Q1)- add a diagnostic flow-chart, especially to highlight the diagnoses of exclusions (mimicking TORCH).

A1) The flow chart of diagnosis and differential diagnosis added (Fig.1).

Q2) - Table 1 shows some laboratory tests. Highlight in bold those specific for prognostic characterization. Highlight with an asterisk those that represent a differential diagnosis of exclusion. Add a legend to the table to describe everything

A2) The changes done, the legend added.

Q3) The role of the laboratory is not clear from the text. Do the tests presented guide and confirm the diagnosis?

A3) The information about lab tst were added in the text.

Q4) Comments on the Quality of English Language

If possible, revise the text to improve reading fluency.

A4) The English editing done.

Thank you so much!

On behalf of the Authors

Mikhail Kostik MD, Ph.D, Professor